Aggressive signaling among competing species of birds

Kenyon Haley L. haley.kenyon@queensu.ca
http://orcid.org/0000-0003-2611-1211 Martin Paul R.
Department of Biology, Queen’s University , Kingston, Ontario , Canada
Curley James
Electronic publication date: 2022 Jun 13
Publication date: 2022
Volume: 10
Electronic Location ID: e13431
Received 2021 Jun 14; Accepted 2022 Apr 21
Copyright: © 2022 Kenyon and Martin
Copyright year: 2022
Copyright holder: Kenyon and Martin
License: This is an open access article distributed under the terms of the Creative Commons Attribution License, which permits unrestricted use, distribution, reproduction and adaptation in any medium and for any purpose provided that it is properly attributed. For attribution, the original author(s), title, publication source (PeerJ) and either DOI or URL of the article must be cited.
License URL: https://creativecommons.org/licenses/by/4.0/

Keywords: Interspecific aggression, Signals, Agonistic interactions, Dominance hierarchies, Competition, Fighting, Color badges

Funding: Natural Sciences and Engineering Research Council of Canada RGPIN/04452-2018 and CGSD3-476023-2015 This work was funded by a Natural Sciences and Engineering Research Council of Canada grant to Paul R. Martin (RGPIN/04452-2018) and fellowship to Haley L. Kenyon (CGSD3-476023-2015). The funders had no role in study design, data collection and analysis, decision to publish, or preparation of the manuscript.

==============================
Aggressive interactions help individuals to gain access to and defend resources, but they can be costly, leading to increased predation risk, injury, or death. Signals involving sounds and color can allow birds to avoid the costs of intraspecific aggressive encounters, but we know less about agonistic signaling between species, where fights can be frequent and just as costly. Here, we review photographic and video evidence of aggressive interactions among species of birds (N = 337 interactions documenting the aggressive signals of 164 different bird species from 120 genera, 50 families, and 24 orders) to document how individuals signal in aggressive encounters among species, and explore whether these visual signals are similar to those used in aggressive encounters with conspecifics. Despite the diversity of birds examined, most aggressively signaling birds displayed weapons (bills, talons, wings) used in fighting and placed these weapons closest to their heterospecific opponent when signaling. Most species oriented their bodies and heads forward with their bills pointing towards their heterospecific opponent, often highlighting their face, throat, mouth, and bill. Many birds also opened their wings and/or tails, increasing their apparent size in displays, consistent with the importance of body size in determining behavioral dominance among species. Aggressive postures were often similar across species and taxonomic families. Exceptions included Accipitridae and Falconidae, which often highlighted their talons in the air, Columbidae, which often highlighted their underwings from the side, and Trochilidae, which often hovered upright in the air and pointed their fanned tail downward. Most species highlighted bright carotenoid-based colors in their signals, but highlighted colors varied across species and often involved multiple colors in combination (e.g., black, white, and carotenoid-based colors). Finally, birds tended to use the same visual signals in aggressive encounters with heterospecifics that they use in aggressive encounters with conspecifics, suggesting that selection from aggressive interactions may act on the same signaling traits regardless of competitor identity.

Introduction

Aggressive behaviors help individuals to gain access to and defend resources such as food, territories, mates, nesting sites, display sites, and roosting sites. These behaviors, however, can be costly when aggressive encounters escalate to physical battles, which can be energetically demanding (e.g., Riechert, 1988; Rovero et al., 2000; deCarvalho, Watson & Field, 2004; Briffa & Sneddon, 2007; Viera et al., 2011) and result in increased predation risk (e.g., Jakobsson, Brick & Kullberg, 1995; Diniz, 2020), injury (e.g., Robertson, Gibbs & Stuchbury, 1986), or death (e.g., Hof & Hazlett, 2012; Lowney et al., 2017; Guo & Dukas, 2020). Individuals can settle disputes without incurring these costs by instead signaling during aggressive encounters. Such signals commonly broadcast aggressive intent (e.g., Husak, 2004; Van Dyk & Evans, 2008; Kareklas, McMurray & Arnott, 2019), fighting ability (e.g., Clutton-Brock & Albon, 1979; Arnott & Elwood, 2009; Palaoro & Peixoto, 2021), and dominance status (Senar, 2006), allowing competitors to assess their chances of winning a physical battle, and thus resolve a dispute, while minimizing risk.

In birds, signals used in aggressive encounters between conspecifics have been well-studied and some generalizations can be drawn. Vocal signals can play a key role in agonistic interactions; specific changes in song can signal step-wise increases in aggressive intent (Searcy & Beecher, 2009), and many species signal an impending physical attack by singing soft songs as they approach a competitor (e.g., Dabelsteen et al., 1998; Akçay et al., 2015). In addition, coloration is often an honest signal of dominance status among conspecific birds, with dominant individuals in many species having larger, and sometimes more intense, badges of status (Senar, 2006). These badges often involve melanin pigmentation, but other types of coloration are also used, depending on the species (Senar, 2006; Santos, Scheck & Nakagawa, 2011).

Interference competition, however, does not just occur between conspecifics; aggressive encounters frequently involve members of different species that compete for shared and limiting resources (Peiman & Robinson, 2010; Fig. 1). Like intraspecific disputes, these interactions can be costly (e.g., Livezey & Humphrey, 1985; Nuechterlein & Storer, 1985; Potti et al., 2021), suggesting that selection should favor the use of signals among heterospecifics during aggressive contests, but the signals used among competing species remain less explored (Caro & Allen, 2017). Some studies, however, suggest that vocalizations (e.g., Gorton, 1977; Catchpole, 1978; Rice, 1978; Reed, 1982; Martin et al., 1996; Martin & Martin, 2001; Jankowski, Robinson & Levey, 2010; Sosa-López, Mennill & Renton, 2017) and color (e.g., Flack, 1976; König, 1983; Snow & Snow, 1984) may signal aggression or dominance in competitive contests among species. Furthermore, several species of birds appear to signal their subordinance to avoid heterospecific aggression from co-occurring dominant species (Gill, 1971; Sætre, Král & Bičík, 1993). These examples lead to the general question: what signals do birds use in aggressive contests among species, and how do these signals vary across diverse groups of birds?

Figure 1 A male Red-winged Blackbird (Agelaius phoeniceus) signals aggressively at a Blue Jay (Cyanocitta cristata) at a bird feeder.

The study of signaling in Red-winged Blackbirds, and their use of red epaulets, has centered on intraspecific function (Smith, 1972; Røskaft & Rohwer, 1987; Yasukawa & Searcy, 2020); however, blackbirds often direct their aggressive displays towards heterospecifics in competitive interactions. Image is a still shot from a Cornell Lab of Ornithology Bird Cam video, available from https://www.youtube.com/watch?v=8QUZEBgeMPk, and reproduced with permission from the Cornell Lab of Ornithology Bird Cams (www.AllAboutBirds.org/Cams). This interaction is an example of one of the interactions included in our dataset.

Here, we review photographic and video evidence of aggressive interactions among species of birds contesting a resource to describe the aggressive signals used by different species and taxonomic groups. Specifically, we reviewed photographs and video of aggressive signaling among species, and used this evidence to (1) identify postures, body regions, and colors used to signal aggression towards competing species, and (2) compare these postures and body regions to those used to signal aggression towards conspecifics. For groups with sufficient sample sizes, we also used this evidence to assess variation in postures and body regions used by different individuals (3) within species, and (4) among taxonomic families of birds.

Materials and Methods

Signal data

We compiled published videos and photographs of aggressive signaling between different species of birds in the context of competition or defense of resources. The majority of the videos and photographs (N = 259) that we used were available from WikiAves (https://www.wikiaves.com.br/), a Brazil-based website for birdwatching and citizen science, where users can publish photos and recordings in a searchable database focused on Brazil and nearby countries (since 2008). Our goal was to find material documenting the interspecific signals used by diverse groups of birds, and thus we prioritized material representing diverse taxonomic families; the avian biodiversity of South America represented in the WikiAves database provided an excellent foundation for this dataset.

We relied on descriptions of the context of the videos or photographs to ensure that we included only media capturing aggressive encounters in our dataset. In WikiAves, we used the “Advanced Search” tool, filtering under “Main Action” to include only “fighting” interactions, and then used only the resulting images involving interactions among multiple species (database searched July 2019). Similarly, we searched the Internet Bird Collection database (HBW.com, searched July 2019, now available in The Cornell Lab of Ornithology Macaulay Library: macaulaylibrary.org) for media with captions including the word “fight*” and included only results depicting interspecific interactions in our study. We note that the Macaulay Library – an excellent source for photographs and video – did not have the advanced search options to identify fighting birds, and thus we were unable to search this data source for our study. We included additional images of interspecific aggressive interactions from other sources (e.g., Martin & Briskie, 2021; YouTube.com: searched July 2019 for specific taxa and the search term “fight”) for certain groups that were underrepresented in our dataset to increase the taxonomic breadth of our study. Sources and credit for each item are listed in our data set.

We further refined our dataset to focus only on signals used in aggressive interactions between heterospecifics. We did not include photographs or video segments that depicted direct fights (i.e., physical contact between competitors) or chases, and instead focused on displays that did not involve contact between focal individuals. We note that signals used during direct fights and chases were thus omitted from our study, and could be different from those used in aggressive signaling without contact. In addition, we did not include birds that were retreating from a resource or interaction in our dataset because we wished to characterize aggressive signals; retreating from a resource or interaction meant that the focal bird was no longer aggressively challenging other species. Retreating birds may signal submission to aggressive heterospecifics (see also Gill, 1971; Sætre, Král & Bičík, 1993); however, our study was not designed to identify such signals. We considered a bird as retreating from a resource or interaction when it appeared to be actively moving away from the resource or aggressively signaling heterospecific, respectively (e.g., beginning to fly or move away). For photographs, we used only images that appeared to capture a full display or signal, although we could not always be sure that the image captured the point of peak intensity.

Our dataset is comprised of photographs and videos (N = 337 interactions) of 164 different bird species from 120 genera, 50 families, and 24 orders, following the IOC World Bird List taxonomy (Gill, Donsker & Rasmussen, 2020). It includes a diversity of aggressive interactions, with birds signaling from a perch (53%), the ground (16%), the air (22%), and the water (9%). While interactions from South America comprise the majority of our dataset (N = 259), we also include interactions from Africa (N = 17), Europe (N = 15), North America (N = 14), Oceania (N = 12), Asia (N = 9), and Antarctica (N = 7). For each aggressive encounter, we documented details of the signaler’s posture, and the body regions and colors that the signaler highlighted.

Postures

We examined the postures used during aggressive encounters with heterospecifics by categorizing the posture of each focal bird in our dataset focusing on eight different components (Table 1). We categorized the overall body orientation and the position of six different body regions: the head, wing, shoulder, bill, tail, and feet. We also recorded which part of the signaler’s body was closest to the receiver (i.e., the heterospecific individual to which the focal individual was signaling). We define each category of posture with photo examples in Table SI.

Table 1 Categories used to describe the position of eight different components of a bird’s posture.

We define each category of posture with photo examples in Table SI.

Posture component	Definition	Position categories	
Body orientation	Overall orientation of the signaler’s body	Forward-upright
Forward-lowered
Forward-normal
Side-oriented
Feet-forward
Above the other species
Upside-down	
Head position	Position of the signaler’s head relative to their body	Forward-upright
Forward-lowered
Forward-normal
Side-oriented
Held-back-and-upwards	
Wing position	Position of the signaler’s wings relative to their body	Flapping (for birds actively flapping or hovering in the air)
Soaring-gliding
Spread-outward
Raised-upward
Partially-spread
Closed-flat
Closed-held-slightly-out
Closed-raised-off-back	
Shoulder position	Position of the shoulders (including underwing/upperwing) relative to the receiver	Underwing-forward
Upperwing-forward
Wing-horizontal (i.e., shoulder forward, flight feathers trailing)
Wing closed with shoulder visible
Wing closed with shoulder concealed	
Bill position	Position of the bill relative to the receiver, and whether it is open or closed	Open-forward
Open-upward
Open-downward
Open-side
Closed-forward
Closed-upward
Closed-downward
Closed-side
Note: For video segments, we recorded the bill as ’open’ if it was opened at some point during the aggressive signaling.	
Tail position	Position of the tail relative to the body and receiver, and whether or not the tail was fanned	Trailing-fanned
Trailing-not fanned
Raised-fanned
Raised-not fanned
Down-fanned
Down-not fanned
Partly raised-fanned
Partly raised-not fanned
Side-oriented-fanned
Side-oriented-not fanned	
Feet position	Position of the feet	On-substrate (including ground, water or perch)
Tucked-up
Extended
Partially extended
Hanging	
Closest point	Closest part of the signaler’s body to the receiver	See body regions illustrated in Fig. 2.	

Body regions highlighted by signaling birds

We identified body regions that focal individuals featured most prominently (i.e., highlighted) in aggressive displays, grouping regions by location on the bird and their likelihood of being collectively visible. Some regions that are typically examined separately in studies of coloration (e.g., lores, forehead, auriculars, chin, throat) tended to be visible together, and thus we grouped them for our analyses (e.g., face/throat). We illustrate the regions included in our study in Fig. 2. Distinguishing between these highlighted regions is somewhat subjective; thus, each photograph or video was examined separately by three naïve human observers, who each recorded up to three regions that they perceived as most highlighted by the signaler from the positional perspective of the heterospecific receiver (Appendix II). For our analyses, we considered a body region to be highlighted in an interaction if it was indicated as highlighted by at least two of the three observers.

Figure 2 Body regions included in our study.

Paintings by Paul R. Martin.

Colors highlighted by signaling birds

We summarized the colors used in interspecific aggressive signaling by recording the colors that focal birds most commonly highlighted in aggressive signals towards other species. The three naïve observers recorded which three colors or color groups were most prominently featured (i.e., highlighted) by the signaler in each interaction from the positional perspective of the heterospecific receiver, categorizing colors or color groups as: carotenoid (red/orange/pink/yellow), structural (blue/green/violet), black, white, rufous/chestnut, brown/beige, gray, or contrasting black/dark and white (Appendix II). For our analyses, we considered a color or color group to be highlighted in an interaction if it was indicated as highlighted by at least two of the three observers. We note that in most signals, these are the colors of the body regions that were identified as highlighted, or parts thereof, but in others these may be the colors of different body regions or a single body region.

To differentiate between colors that were present in a signaling species, but were not highlighted, and those that were not present, we consulted written descriptions of the coloration of each species in our study. We relied on color descriptions from Birds of the World (Billerman et al., 2020) for all species, except for one species (Leptoptilos crumenifer), which was missing a complete color description in this source; we thus consulted another source for a complete color description for this species (Neudamm, 1900). References for the color descriptions that we used for all species in our study are available in our dataset. For each species, focal colors or color groups that are present, but not highlighted in a focal interaction, are indicated by 0 in our dataset, and focal colors or color groups that are not present in the focal species are indicated by NA, and thereby excluded from the analysis. Appendix III provides more information about the color names considered to be part of each focal color category.

Relationship to intraspecific signals

We addressed whether interspecific aggressive signals differed from signals used in intraspecific aggression by comparing the characteristics of interspecific aggressive signals described above (postures, focal regions for signaling) to the characteristics of signals used in within-species aggressive interactions. We did not incorporate color into these comparisons because some species have uniform coloration and thus color is uninformative with respect to patterns of within- vs. among-species signaling. For species already included in our dataset, we found information on intraspecific aggressive signals in published photographs and videos of aggressive interactions (N = 141 species). For these species, we also incorporated descriptions of intraspecific aggressive signals from the literature, where available (i.e., for better-studied species: N = 20 of the 141 species included in this component of the study). We described the degree of similarity between interspecific and intraspecific aggressive signals as the ‘same’ if the same postures and highlighted regions were used for both intraspecific and interspecific signaling, ‘different’ if most (>50%) of the postures and highlighted regions differed, and ‘similar’ if some, but not most, postures, highlighted regions differed. We were unable to obtain information about intraspecific aggressive signals for some species in our dataset (N = 36 interactions); for these species we compared observed signals to intraspecific aggressive signals used by their congeners, where possible (N = 6 interactions were designated as ‘similar to congener’).

Statistical tests

We conducted all of our statistical analyses and plotting in R (R Core Team, 2020). We provide the R code that we used for our analyses and figures, along with our dataset, as Supporting Information.

We used Bayesian phylogenetic mixed-effects models using the MCMCglmm function in the MCMCglmm R package (version 2.32; Hadfield, 2010) to test whether birds were more likely to use specific postures, or highlight specific body regions or colors in aggressive encounters with heterospecifics, as well as to compare whether birds use the same signals in interspecific aggressive interactions as in intraspecific aggressive encounters. Bayesian phylogenetic mixed-effects models allowed us to incorporate the effects of phylogeny in our analyses. We obtained a phylogeny for the signaling species in our dataset from Jetz et al. (2012) (birdtree.org; maximum clade credibility tree of 1,000 Hackett all-species trees; Fig. S1) and included the phylogeny, species, and focal.interaction (i.e., a numeric identifier for the documented interaction) as random effects in each model. We did not specify priors for fixed effects so that all variance parameters were estimated (Hadfield, 2018), but specified priors where V = 1 and nu = 1 for both R and G structures (Hadfield, 2010, 2018). We used a categorical (i.e., binomial) distribution and ran simulations for 2,000,000 iterations (burnin = 10,000 iterations, thinning interval = 100) for all analyses, except for two posture analyses examining tail position and the part of the signaler’s body positioned closest to the receiver, both of which had position categories that were rarely assumed and were thus run for more iterations (5,000,000 iterations) to achieve sufficient effective sample sizes. For each analysis, we tested whether the model with the focal predictor performed better than a null model using DIC values (better performing models were identified as those with lower DIC values; Spiegelhalter et al., 2002), thus indicating that our focal birds differentially used postures, or differentially highlighted body regions or colors, in aggressive encounters with heterospecifics. We provide details about model diagnostics in the Supporting Information.

We used a separate model for each component of posture (i.e., body orientation, head position, etc.) to test whether different positions of each component (e.g., forward.lowered, forward.upright, etc.) were equally likely to be assumed in interspecific aggressive interactions. We used a binary response variable indicating whether each position was assumed (1 = position assumed; 0 = position not assumed) as the response variable, and position as a categorical predictor variable. To describe the body regions most commonly highlighted in aggressive signals, we calculated the proportion of interactions in our dataset which highlighted each body region (as indicated by at least 2/3 naïve observers). To test whether certain body regions were most likely to be highlighted, we used a binary response variable indicating whether the focal body region was highlighted (1 = highlighted; 0 = not highlighted) as the response variable and body region as a categorical predictor variable. We excluded three body regions from our analysis (uppertail coverts, undertail coverts, and tarsal feathers) because they were never identified among the most highlighted body regions in our dataset.

Similarly, to describe the color or color groups most commonly highlighted in aggressive signals, we calculated the proportion of interactions in our dataset in which the signaler highlighted each color or color group (as indicated by at least 2/3 naïve observers). To test whether certain colors or color groups were most likely to be highlighted, we used a binary response variable indicating whether the signaler highlighted the focal color (1 = highlighted; 0 = not highlighted) as the response variable and color as a categorical predictor variable. Colors or color groups that were not present in the signaling species in each focal interaction were excluded from this analysis.

To understand whether birds use the same signals in aggressive encounters with heterospecifics as in aggressive encounters with members of their own species, we calculated the proportion of interspecific interactions in our dataset in which the signal used had each degree of similarity to intraspecific signals (same, similar, similar to congener, or different). We used a binary response variable indicating whether each degree of similarity was assumed (1 = yes; 0 = no) as the response variable and degree of similarity as a categorical predictor variable.

To ensure that source format (photo vs. video) did not have a large influence on our results, we repeated all MCMCglmm models with the subset of the dataset captured in photographs (N = 289 interactions; see Appendices VI, VIII, and IX in Supporting Information).

We used binomial tests using the binom.test R function to examine the consistency of signals used in aggressive encounters within a species. For species for which we had sufficient data (>6 videos or images) we tested whether the majority of birds in a species (>50%) used certain postures or highlighted certain body regions during aggressive encounters with heterospecifics.

Finally, to examine how the signals used in interspecific aggressive interactions vary among families, we calculated the proportions of each family that used each posture, and highlighted each body region and color. We included only families that were sufficiently represented (>6 videos or images) in our dataset in this analysis (N = 14 families).

Results

Postures assumed, and body regions and colors highlighted during aggressive encounters with heterospecifics

The model that included the focal predictor performed better than the null model in all of our analyses (Tables S3, S14), indicating that our focal birds differentially used postures, and differentially highlighted body regions and colors in aggressive encounters with heterospecifics.

Each body region position was not equally likely to be used in aggressive signals by birds in our dataset (Fig. 3; Table S3). During aggressive interactions with other species, birds in our dataset more commonly assumed a forward-facing, lowered body position (42% of aggressive interactions in our dataset; Fig. 3A; Table S4), with a forward-facing, lowered head (52%; Fig. 3B; Table S5), open wings (57%; Fig. 3C; Table S6), open, forward-facing bills (68%; Fig. 3E; Table S8), and trailing, unfanned tails (39%; Fig. 3F; Table S9). Birds were more likely to hold their bill closest to the heterospecific receiver than other body regions (87%; Fig. 3H; Table S11). Birds in our dataset were more likely to plant their feet on a substrate during aggressive signals than to assume other foot positions (82%; Fig. 3G; Table S10), and they were equally likely to assume a shoulder position with their underwing forward (38%; Fig. 3D), as a shoulder position with their wing closed with their shoulder visible (34%; Fig. 3D), but other shoulder positions were less common (Fig. 3D; Table S7).

Figure 3 Postures assumed during aggressive encounters with heterospecifics.

(A) Body orientation (N = 337), (B) head position (N = 336), (C) wing position (N = 337), (D) shoulder position (N = 336), (E) bill position (N = 316), (F) tail position (N = 295), (G) feet position (N = 326), and (H) closest point to receiver (N = 337). Gray circles show the raw data jittered (1 = posture assumed in focal interaction; 0 = posture not assumed in focal interaction), black points show model estimates back-transformed from log odds, and error bars show 95% credible intervals. Letters in each panel indicate differences between estimates for each posture category; estimates with the same letter are not significantly different from one another.

Birds in our study were more likely to highlight their face and throat area than other body regions during aggressive signals directed towards heterospecifics (77% of aggressive interactions in our dataset; Fig. 4A; Table S15). Birds also more commonly highlighted their mouth (37%; Fig. 4A; Table S15), underwings (27%; Fig. 4A; Table S15), and bill (27%; Fig. 4A; Table S15) than other body regions, including the breast, nape/back, legs/feet, crown, undertail, upperwings, belly, uppertail, shoulders, and sides. No birds in our dataset highlighted their uppertail coverts, undertail coverts, or tarsal feathers in aggressive interactions with members of other species (Fig. 4A). Birds were more likely to highlight carotenoid colors, including red, pink, orange, and yellow, than other colors in the aggressive signals captured in our study (55% of aggressive interactions in our dataset; Fig. 4B; Table S16).

Figure 4 Body regions and colors highlighted in aggressive interactions with heterospecifics.

(A) The face and throat region was the most likely body region to be highlighted in aggressive signals (N = 337). (B) Carotenoid colors (red, pink, orange, or yellow) were the most likely color group to be highlighted in aggressive signals (carotenoids: N = 316; blue/green/violet: N = 189; dark-white contrast: N = 294; black: N = 329; brown/beige: N = 303; gray: N = 308; white: N = 299; rufous/chestnut: N = 132). Gray circles show the jittered raw data (1 = highlighted; 0 = not highlighted), black points show model estimates back-transformed from log odds, and error bars show 95% credible intervals. Letters in each panel indicate differences between estimates for categorical predictors; estimates with the same letter are not significantly different from one another.

Model results for postures, highlighted body regions, and highlighted colors remained similar when run using only the subset of the data obtained from photographs (Appendices VI, VIII), indicating that the source format had little influence on our results.

Similarity to signals used in aggressive encounters with conspecifics

We scored the similarity of 307 of the interactions in our dataset to signals used during aggressive encounters with conspecifics. The model that included the degree of similarity as a predictor performed better than the null model (Table S20). The majority of birds in our dataset used the same signal (body position and highlighted body regions) during aggressive encounters with heterospecifics that they use in aggressive encounters with conspecifics (87% of aggressive interactions in our dataset; Table S21). Model results remained similar when run using only the subset of the data obtained from photographs (Appendix X), indicating that the source format had little influence on our results.

Within-species similarity in signals used in aggressive encounters with heterospecifics

Four species were sufficiently represented in our study to be examined for consistency in the signals that they use during aggressive encounters with heterospecifics. Each of these species had several posture categories that remained consistent across all images or videos. Columbina talpacoti (Columbidae) always had their wings raised upward and their feet planted on the substrate (N = 7, binomial test: P = 0.008). Eupetomena macroura (Trochilidae) always had a forward, upright body position, a fanned tail pointed down, and held their bill closest to the heterospecific competitor (N = 7, binomial test: P = 0.008). Similarly, the bill of Pitangus sulphuratus (Tyrannidae) was always one of the body parts closest to the other bird (N = 13, binomial test: P < 0.001). Thraupis sayaca (Thraupidae) always had an open, forward-facing bill position, planted feet, and again, held their bill closest to the heterospecific competitor (N = 24, binomial test: P < 0.001).

All four species highlighted at least one body region in the majority of signals in our dataset. All Columbina talpacoti individuals highlighted their underwings (N = 7, binomial test: P = 0.008). All Eupetomena macroura highlighted their face/throat (N = 7, binomial test: P = 0.008). The majority of Pitangus sulphuratus individuals in our dataset highlighted their face/throat region (77%; N = 13, binomial test: P = 0.046). Similarly, Thraupis sayaca individuals in our dataset tended to highlight their face/throat (96%; N = 24, binomial test: P < 0.001) and mouth (75%; N = 24, binomial test: P = 0.01).

Within- and among-family variation in signals used in aggressive encounters with heterospecifics

Many birds in our study showed some aspects of their posture that were fairly stereotyped across interspecific aggressive signals by members of their family, and some postures were fairly unique to a specific family (Fig. 5). Members of Falconidae always assumed an underwing.forward shoulder position and typically extended their feet (92%; Table S24). Anatidae and Spheniscidae always positioned their bill forward and open, and had a trailing, unfanned tail (Table S24). Fringillidae and Turdidae also always held their bill open and forward (Table S24). Columbidae was the only family in which most birds in our dataset did not have a forward-oriented body orientation (75% side oriented; Table S24) or head position (58% side oriented; Table S24). Columbidae, Falconidae, and Trochilidae always had open wing positions (i.e., flapping, spread.outward, partially.spread, raised.upward, soaring.gliding; Fig. 5; Table S24). While most families in our study positioned their bill closest to their competitor, the bill was not involved in the majority of interspecific aggressive signals by members of Accipitridae (50% feet; Table S24), Columbidae (50% wing; Table S24) or Falconidae (86% feet; Table S24).

Figure 5 Variation in the most common postures used in aggressive signaling towards heterospecifics across the 14 focal families examined in this study.

In most families, signaling birds direct their face and point their bill towards the heterospecific opponent. In Columbidae, signaling birds typically line up sideways, with their underwing closest to the heterospecific opponent. In Accipitridae and Falconidae, signaling birds typically extend their legs so that their talons are closest to the heterospecific opponent. Accipitridae, Trochilidae, Falconidae, and Tynrannidae most commonly signaled in the air, while the rest of the families most common signaled from the ground or water. Illustrated species are: Tadorna tadorna (Anatidae), Eupetomena macroura (Trochilidae), Columbina talpacoti (Columbidae), Pygoscelis papua (Spheniscidae), Diomedea antipodensis gibsoni (Diomedeidae), Ardea goliath (Ardeidae), Haliaeetus pelagicus (Accipitridae), Falco femoralis (Falconidae), Megarynchus pitangua (Tyrannidae), Turdus merula (Turdidae), Euphonia chalybea (Fringillidae), Tangara (Thraupis) cyanoptera (Thraupidae), Icterus pyrrhopterus tibialis (Icteridae), Eupsittula aurea (Psittacidae). Paintings illustrate postures from photos and video of interactions. Paintings by Paul R. Martin.

Most families in our dataset showed intra-family variation in which body regions they were most likely to highlight in interspecific aggressive signals (Table S25). Members of Anatidae, however, always highlighted the mouth, while members of Ardeidae, Columbidae, Falconidae, and Icteridae never did so (Fig. 6; Table S25). Falconidae was the only family in which most members were most likely to highlight their legs and feet (Figs. 5, 6; Table S25).

Figure 6 Phylogeny of all species in our dataset and the body regions most commonly highlighted by each species.

Each column corresponds to one body region: black bars in a column indicate a region that is highlighted by the focal species in at least half of the interactions in our dataset in which the focal species is the signaler; bars filled with light gray indicate a region that is not highlighted by the focal species (i.e., highlighted in less than half of the interactions in our dataset in which the focal species is the signaler). Numbers along the x-axis are branch lengths (Mya).

Many colors or color groups were highlighted by many different families included in our study, but we observed some differences among families in the signals commonly used in aggressive interactions with heterospecifics (Fig. 7; Table S26). In our dataset members of Diomedeidae always highlighted white and members of Spheniscidae always highlighted contrasting dark and white in aggressive signals. Despite carotenoid colors being commonly highlighted by most families in our dataset, Columbidae never highlighted red, pink, orange, or yellow, and instead were most likely to highlight gray in aggressive interspecific interactions (Fig. 7; Table S26). We note that no families that were widely represented in our study most commonly highlighted rufous or chestnut, while all families had some representatives that highlighted black (Fig. 7; Table S26).

Figure 7 Phylogeny of all species in our dataset and the colors most likely to be highlighted by each species.

Each column corresponds to one color group: colored bars in a column indicate a color/color group that is highlighted by the focal species in at least half of the interactions in our dataset in which the focal species is the signaler (red for red.pink.orange.yellow, blue for blue.green.violet, black and white stripes for dark.white.contrast, black for black, brown for brown.beige, dark gray for gray, white with black borders for white, rufous for rufous or chestnut); bars filled with light gray indicate a color/color group that is not highlighted by the focal species (i.e., highlighted in less than half of the interactions in our dataset in which the focal species is the signaler); blank spaces indicate colors that are not present in the focal species. Numbers along the x-axis are branch lengths (Mya).

Discussion

Aggressive interactions among competing species can be costly and dangerous (e.g., Livezey & Humphrey, 1985; Nuechterlein & Storer, 1985; Potti et al., 2021), thus favoring the use of signals that allow individuals to avoid the risk of physical fights with other species (Caro & Allen, 2017). Here, we used publicly available videos and photographs of aggressive encounters between different species of birds to examine the aggressive signals used by different species and taxonomic groups, describing the postures used, and body regions and colors commonly highlighted during these interactions. Our study includes 164 different bird species from 121 genera, 50 families, and 24 different orders to show broad similarities and key differences in signals directed at competing heterospecifics.

Despite the diversity of taxa examined, most species of birds in our study highlighted weapons used in fighting (bill, talons, wings) (Fig. 4A), and held these weapons closest to their heterospecific opponent (Fig. 3H). In most species, the bill was directed at the opponent (i.e., oriented forward and held closest to the opponent), either open (e.g., Anatidae) or closed (e.g., Trochilidae) (Fig. 3). We note that the bill may be the closest point to a competitor when a bird is in a neutral position but facing a competitor; however, many birds in our study assumed a forward, lowered body position, which thrusts the bill closer to a competitor (Fig. 3). For species with well-developed talons used in hunting (Accipitridae, Falconidae), the legs, rather than bills, were typically extended outward towards the opponent, with talons opened. These displays required that the birds be positioned in the air; falcons typically achieved these signals while flying with talons dangling, while hawks and related birds either extended their talons while flying, or threw them up towards the opponent from the ground, a perch, or as they approached in the air. Pigeons and doves (Columbidae) stood out as an exception among most birds; they typically lined up sideways to their opponents with one or both wings raised, an underwing closest to the other species. This posture aligns with their specialized means of fighting, where they often pound opponents with their wings (e.g., Otis et al., 2020). Species possessing other specialized weaponry, such as bony spurs, often highlighted the body regions where these weapons are found in aggressive interactions with heterospecifics; for example, species with bony spurs at their carpal joints (genera Vanellus and Jacana, here) displayed them by highlighting their underwings.

Across birds, many species also spread their wings (e.g., Ardeidae, Diomedeidae) and sometimes their tails (e.g., Trochilidae, Falconidae), augmenting their apparent size to their opponents. Body size is the best predictor of behavioral dominance among aggressively competing species of birds and other animals (Morse, 1974; Peters, 1983; Robinson & Terborgh, 1995; Donadio & Buskirk, 2006; Martin & Ghalambor, 2014; Miller et al., 2017), and thus extending wings and tails to highlight, or even exaggerate, size could provide an important signal of dominance and threat. Spread wings or tails were not components of the most common signals across all families of birds (Fig. 5), but these postures were observed in most families in our dataset. Several species also showed striking color patterns associated with extended wings (e.g., bright or contrasting underwing coverts: Ardeidae, Columbidae; eye spots on upperwings, Eurypygidae). Hummingbirds typically signaled in the air with wings moving extremely rapidly; these species most often faced their opponents with fanned tails that exhibited striking color patterns, shapes, and plumes.

The highlighting of similar traits (weapons, size) in aggressive signals among diverse species of birds makes sense from the perspective of the evolution of signals to convey information to other species. A signal of aggression towards another species would be most effective if it could be easily understood by any competitor; a species-specific signal, on the other hand, would require that heterospecifics learn the information conveyed in the signal through costly aggressive contests. Weaponry and size provide information about aggressive intent and fighting ability within most species, and thus serve as a ‘shared language’ when used as a signal towards other species, including distantly-related competitors such as mammals (Kruuk, 1967).

Few families in our study used postures in aggressive displays that were unique relative to other families (Fig. 5). This is in part due to the similarities in many aggressive displays across diverse families of birds (Figs. 6 and 7), which suggests that our results may be broadly applied. Only two families commonly used postures that were rarely seen in other families (i.e., where over 70% of individuals using a specific combination of positions were members of the same family). Trochilidae (hummingbirds) often displayed with open or rapidly flapping wings, with a forward orientation, downward facing tails, and closed bills pointing at their opponents. Columbidae (pigeons and doves) often displayed with raised wings and a side orientation. The body regions most highlighted in aggressive displays were also variable within and among families, with similarities among many families (Table S25). Accipitridae and Falconidae, however, were the two families most likely to highlight their legs and feet during displays (Table S25).

Carotenoid colors were most commonly highlighted in aggressive encounters with heterospecifics across all species of birds, when they were present (Fig. 4B). This may suggest that the use of carotenoids to signal dominance or quality in intraspecific interactions seen in some groups (Senar, 2006) extends to interspecific interactions. The colors highlighted, however, varied within and among taxonomic families, with some families being instead more likely to highlight white (e.g., Diomedeidae), black (e.g., Icteridae), or contrasting combinations of dark and white colors (e.g., Spheniscidae) (Figs. 5 and 7, Table S26). Birds commonly highlighted multiple colors in aggressive displays, including rich or warm colors with black and white patches, all in combination (e.g., Figs. 5 and 7).

Aggressive display postures and highlighted colors were fairly consistent within species, with variation in components of the display perhaps reflecting varying intensities of interactions. Even within an extended interaction between two individuals, the displays varied as the birds interacted. For example, video of extended interactions shows birds consistently directing their bills and faces towards the opponent, but their head positions often vary throughout the interaction, as does whether their bills and wings are open (e.g., cranes, https://macaulaylibrary.org/asset/201385401).

For most species, aggressive displays towards heterospecific competitors were similar to those directed towards conspecifics (see also Martin & Briskie, 2021 for Diomedeidae). This suggests that interactions with other competing species act as a selective pressure on the same displays and traits that are typically studied from the perspective of intraspecific function (e.g., Fig. 1). The broad importance of the bill, face and throat in aggressive signaling with other species (Figs. 3H and 4A) is consistent with previous studies (Dow, 1975; Kalinoski, 1975; Flack, 1976), and suggests that colors and patterns that signal dominance should be more likely to evolve in these regions, and less likely to evolve in regions such as uppertail coverts that signalers never highlighted in aggressive displays. Nonetheless, some regions involved in aggressive signaling varied among taxonomic groups, suggesting that different regions should be the target of selection for signaling depending on the group. Consistent with this idea, we find brightly colored feet and legs in many falcons and hawks (Brown & Amadon, 1968), and bold black, white, and rufous colored underwings in many doves (Goodwin, 1970), which highlight these distinct regions in displays. Importantly, we find that certain areas that have been omitted from studies of aggressive signaling (including both those conducted on museum specimens: e.g., Shultz & Burns, 2017; Cooney et al., 2019; and those using other methods: Martin, Montgomerie & Lougheed, 2015; Drury, Cowen & Grether, 2020), such as bright mouth linings and underwing patterns, are, in fact, frequently emphasized in the aggressive displays of many bird species.

While the aggressive signals used towards conspecifics and heterospecifics appeared superficially similar, we still have much to learn about how birds use signals in conflicts with heterospecifics. We have compared visual signals used in conspecific and heterospecific interactions at a fairly coarse scale, but more subtle differences may exist. For example, birds often use vocalizations in aggressive interactions with competing species, but how they use these sounds can differ from conspecific interactions. Some birds appear to alter their songs or calls to match or mimic the opposing species (Dobkin, 1979; Veerman, 1994; Gorissen, Gorissen & Eens, 2006; Wilson & Scantlebury, 2006), while other species will alter the timing of their singing to sing overtop of the songs of subordinate species – a behavior that is not used when presented with conspecific songs (Martin & Martin, 2001). Whether birds similarly alter their visual signals in response to competing heterospecifics, perhaps by modifying the frequency or intensity of their displays, remains to be explored.

Our study would not have been possible without the resources provided by birdwatchers and community scientists, and in particular, the Brazilian online resource WikiAves. While aggressive interactions among species are common and important for the structure of ecological communities (Peiman & Robinson, 2010), aggressive signals are fleeting and difficult to observe. Compilations of photos and videos from many independent observers allowed us a unique opportunity to document aggressive signals on a broader taxonomic scale, illustrating the importance of community science datasets for understanding the intersection between behavioral and community ecology.

Conclusions

Our review of aggressive signals among competing species of birds explores similarities and differences among diverse species in the postures, and the body regions and colors that they highlight in aggressive displays. The results suggest that signals used in aggressive contests within species are also used among species, and that aggressive interactions with heterospecifics likely act as a selective pressure on many of the same traits used in within-species interactions. These same traits are often subject to a diverse suite of selective pressures (inter-sexual selection, predation, parasitism), creating synergistic and conflicting pressures that shape their evolution. Given the role of heterospecific aggression and interference competition as important selective pressures on traits (Peiman & Robinson, 2010; Grether et al., 2009, 2013, 2017; Drury, Cowen & Grether, 2020), we hope that future studies of trait evolution will consider the function of signaling traits as mediators of among-species competitive interactions, and recognize their role in dominance interactions and hierarchies among species within communities.

Supplemental Information

Supplemental Information 1 Aggressive signaling among competing species of birds dataset.

Click here for additional data file.

Supplemental Information 2 README file for dataset.

Click here for additional data file.

Supplemental Information 3 Appendices.

Click here for additional data file.

Supplemental Information 4 Phylogeny of species in study for phylogenetic analyses.

Click here for additional data file.

Supplemental Information 5 R script for analyses for Aggressive signaling among competing species of birds.

Click here for additional data file.

We thank E. Basham, K. Esteireiro, J. Niskanen, S. Margorian, and Y. Vangenne for assistance with this project. We thank the photographers whose publicly available photos and videos comprise our dataset; specific credit for each item is listed in our data file. We thank the online resources WikiAves and the Internet Bird Collection for compiling most of the media used in our study and making it publicly available. Computations were performed on computing cluster resources provided by the Centre for Advanced Computing (CAC) at Queen’s University in Kingston, Ontario. The CAC is funded by: the Canada Foundation for Innovation, the Government of Ontario, and Queen’s University. We thank James Curley and two anonymous reviewers for their valuable input.

Additional Information and Declarations

Competing Interests

Author Contributions

Data Availability

The authors declare that they have no competing interests.

Haley L. Kenyon conceived and designed the experiments, performed the experiments, analyzed the data, prepared figures and/or tables, authored or reviewed drafts of the paper, and approved the final draft.

Paul R. Martin conceived and designed the experiments, performed the experiments, prepared figures and/or tables, authored or reviewed drafts of the paper, and approved the final draft.

The following information was supplied regarding data availability:

The code and data, and associated README file, are available in the Supplemental Files.

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
