# Peer review of "Aggressive signaling among competing species of birds"

_PeerJ, doi:10.7717/peerj.13431_

## Round 0.1 · original submission · Major Revisions

· Academic Editor

Major Revisions

Thank you for this interesting submission. The reviewers and I find the work to be very novel and interesting. In terms of the reviews, both reviewers raise a number of questions that should be addressed. In particular, both reviewers have comments regarding the statistical methods that should be carefully considered.

Reviewer 1 ·

Basic reporting

The manuscript is well written. Authors provide a good picture of the aggressive signaling literature; the only reference that I think should have been cited is Caro & Allen (2017, https://doi.org/10.1098/rstb.2016.0344). The manuscript is well structured although it feels a bit convoluted at some points because various analyses were performed without it being clear how each of them relates to the overarching question “what signals do birds use in aggressive contests among species and how do these signals vary across diverse groups of birds?” (particularly the within-species and within-family analyses). Raw data are provided with appropriate sources to back them up. I believe there is an excess of figures, some of which (e.g., Fig. 1, at least some panels in Fig. 2, Fig. 9) could be appended as Supporting Information.

Experimental design

The authors did a great job in identifying and making a case for how interspecific aggressive signaling constitutes a knowledge gap within the field of animal communication. Methods are described with sufficient detail and codes ran smoothly after I corrected a typo (signaling spelled as “signalling” in several instances, causing R to not identify the correct column in the data set). I have numerous concerns about the study design (ranging from data collection through data scoring to data analysis), which I detail in the General comments below.

Validity of the findings

In my view, data are not statistically sound because no analysis accounts for phylogenetic non-independence (see below). I did not inspect conclusions in detail as I have concerns about the methods that generated the findings that should support them in the first place.

Additional comments

In this manuscript, the authors aimed to provide an overview of the signals that birds use during agonistic interactions with heterospecific individuals. To that end, the authors analyzed photos and videos of mostly Neotropical species in aggressive contexts and described several aspects of their gestural displays.

The authors make a good case for why we should give more focus to interspecific aggression signals. The costs of fighting with a heterospecific can be just as dramatic as with a conspecific, so, just like for intraspecific signals, we should expect selection for interspecific signals that can resolve conflicts without physical combats. At the same time, there is reason to expect that interspecific communication should be under distinct selective pressures because of the differences in morphology, perception, and cognition of potential receivers. Since interspecific signals are understudied in comparison with intraspecific signals (a point also made by Caro & Allen 2017, https://doi.org/10.1098/rstb.2016.0344, which the authors should consider citing if they wish to make their case stronger), I believe the authors identified a relevant knowledge gap.

I also commend the authors on coming up with a nice system for classifying the different aspects of gestural displays across species, which ultimately contributes to a very interesting data set. The authors also provide detailed descriptions and illustrated examples for each category in the system, which is of great help to understand how data were scored.

Having said that, I have a number of concerns mostly about how the study was designed, which I will detail below.

I. Statistical approach
I invite the authors to reflect on the following question: why did they decide to use statistical tests?

Statistics are a set of tools that allows us to infer conclusions about a population from a sample of that population. Perhaps the authors are instead interested in making qualitative descriptions pertaining to their sampled species alone, without extrapolating the estimated parameters to the whole population (i.e., all bird species). I say that because the authors do not need a statistical test to be able to state, for example, that “the majority of birds in our dataset had a forward-facing position” (L219). This is a demonstrable fact: in the majority of their sampled interactions, the signaling bird had a forward-facing position. The authors could qualitatively describe the patterns they found for their sampled species (Figures 5–7 seem like a step in that direction) and use them to speculate on how birds in general behave during interspecific aggressive interactions and/or to suggest future directions for the field. That is what non-systematic literature reviews often do.

However, the fact that the authors chose to use statistical analyses tells me that they are interested in extrapolating their findings to birds in general. The problem is that, when we are comparing species, we necessarily have to account for the fact that some species are expected to be more similar to each other because they shared much of their evolutionary histories as a single lineage. In other words, the more recently two lineages diverged from the same branch, the more similar they should be simply because there was not enough time for them to become too different. This makes species non-independent data points that contradict the assumption of independency of traditional statistical tests like the ones used in the manuscript. The problems arising from not incorporating phylogenetic non-independence have been vastly demonstrated since at least Felsenstein (1985, https://doi.org/10.1086/284325) and I am afraid that the authors would necessarily have to account for it if they wish to extrapolate their findings to birds. A nice and thorough overview of phylogenetic comparative methods that do account for phylogenetic non-independence can be found in Garamszegi (2014, https://www.springer.com/gp/book/9783662435496).

If the authors indeed intend to extrapolate their findings to birds in general (with caution, given their small sample size), I suggest that they use phylogenetic generalized linear mixed models (PGLMMs) for all of their analyses. PGLMMs are nothing more than GLMMs that restrict species-specific intercepts to be more similar to each other according to how related species are in the phylogeny. I am not completely familiar with how PGLMMs are implemented in frequentist packages like lme4 but suspect it should be fairly straightforward. A quick search tells me that packages pez and phyr apparently include functions that do so, although I have never used them. Alternatively, function phyloglm in the package phylolm implements phylogenetic logistic regressions, but I am not sure how flexible it is in regard to the other random effects. In lme4-like syntax, the formula would be as follows:

response ~ predictor + (1|interaction id) + (1|species) + (1|phylogeny),

where “response” would be binary (is position assumed? is region/color highlighted? is region closest to receiver? is signal same/similar/different?), “predictor” would be categorical (postures, body regions, colors, signal similarity degree), “interaction id” and “species” would account for repeated measurements from the same interaction and species, respectively (both needed because some interactions involve the same signaling species), and “phylogeny” is the correlation matrix deriving from the phylogenetic tree. The authors can readily sample full bird trees from a Bayesian set available at birdtree.org.

II. Data search
The authors did an excellent job in providing the source for each of the analyzed interactions, but we are given no information on how they searched for data in the first place. Questions that I would like to see answered include: what media/article databases were searched and when? What searching strategies did the authors use? Did they use sets of keywords or did they focus searches on selected species, thereby prioritizing taxonomic diversity (L96)? More importantly, what precise criteria were used to determine whether a picture/video depicted an aggressive display? This last question is important because circular reasoning could be a problem if the criteria are not independent of the scored traits. For example, if open-wings posture, but not closed-wings, was a criterion to consider a behavior as an aggressive display, open wings would surely be well represented in the sample simply because they were used as a criterion, not because they are indeed common as aggressive displays.

III. Color analysis
I have two concerns regarding the color analysis. The first is again related to circular reasoning. In L127, one of the criteria used by the authors to classify a region as highlighted was one “that appeared to highlight contrasting patterns or coloration (e.g., black/white contrasting patches, black throat, bright red crest)”. In other words, the color of a region could determine whether the region was highlighted or not. Later on, naïve observers scored the color of those regions, meaning that colors were used both as a criterion for what constitutes a highlighted region and as a variable that describes that region. If authors automatically considered red, black, and black/white regions as highlighted, they were already biasing the results of the subsequent analysis, which unsurprisingly showed carotenoids, black, and black/white patterns as among the most highlighted colors. My suggestion is that the authors use only the first criterion for what constitutes a highlighted region (“those that were most obviously displayed or modified for display to the receiver”, L125) because it is independent of color. Otherwise, I am afraid that I do not see how the results from this analysis could have much use, unfortunately.

The second issue with this analysis is that colors were scored as 0 (“not highlighted”) even if they were not present at all in a species. An individual can only highlight colors it possesses, which is a sufficient — and biologically uninteresting — explanation for why, for example, albatrosses never highlighted carotenoids or structural colors. It also makes the analysis extremely sensible to the species that ended up in the final sample. For example, one explanation for the fact that rufous/chestnut is rarely highlighted is, say, that birds choose not to highlight this color because it is not recognized as a universal aggressive signal. But an equally likely — and again biologically uninteresting — explanation is that most species in the sample simply did not have any rufous/chestnut in the plumage and therefore could not possibly highlight it. There is no way to distinguish between these two explanations with the way highlighted colors were scored. My suggestion to make this analysis biologically relevant is to score colors as follows: (1) color is present in the plumage and is highlighted; (0) color is present in the plumage and is not highlighted; (NA) color is not present in the plumage. NAs would of course be dropped from the GLMM analysis. By doing as I suggest, 0 will have the unambiguous meaning of “not highlighted” (rather than “either not highlighted or not present”) and thus the logistic regression results will be interpretable as “probability that bird actively highlights a certain color”. (Note that this is not a problem for the highlighted-region analysis because all birds have a face, bill, underwings, breast etc.)

IV. Discussion
I will not get into much detail about the discussion because I would ideally like to see the earlier points resolved before I can turn my eyes to interpretation of the results. But I would like to highlight two points. First, I like the line of reasoning that led the authors to suggest weapons and size augmentation as universal signals. I would be curious to see the result of the new color analysis to see if it follows the same logic (is there a universal color signal?). My only suggestion here is to be more cautious about interpretation of the “closest region” results. In a neutral position, bills are always the closest point to anything birds are facing because they are the most anterior part of the avian body. So, the fact that bills were the part of the body closest to the opponent in the majority of cases can simply mean that the signaling individuals were facing the opponent, rather than actively displaying their bills. Again, I do like the line of reasoning here, but I would appreciate if the authors acknowledged this caveat. My second comment is that paragraphs #5 and 6 (L327–343) are only stating the results in different words, rather than discussing them.

V. Line-by-line comments
L99: could the authors please be more specific about when they considered that a bird was retreating from a resource and why this is relevant for data collection? Perhaps also providing an example?

L146: what was the search strategy for intraspecific signals? Did the authors direct searchers at species for which they already had interspecific data? Did they stop searching after finding a certain number of examples?

L150: could the authors please elaborate on how uniform coloration may obscure differences between intra and interspecific signals? It seems to me that, since the signaling species would be the same, coloration is automatically controlled for in pairwise comparisons between intra and interspecific signals.

L182 and 190: random effect was the same for both GLMM analyses (“Number” variable in the dataset), but described with different terms in the text. I would suggest using the same term for clarity.

L364: it would be interesting to see examples of studies that omit these regions.

L367: I would ask the authors to be more specific on how more subtle differences between intra and interspecific signals could not be picked up by their study.

Reviewer 2 ·

Basic reporting

This paper and its presentation are well-structured, and contains useful figures/tables and also the raw data and code. The introduction is well-presented, but the paper could benefit from a few minor word choice changes to improve clarity, as well as a key word choice change: I encourage the authors to think more carefully about the use of the word “highlighted” and to perhaps come up with a different way to refer to the regions of the body/colors that birds displayed during aggressive interactions.

- Minor wording suggestion: I suggest changing final line of intro (84-87) from “For groups with sufficient sample sizes, we also used this evidence to compare postures and body regions used by different individuals (3) within species, and (4) among taxonomic families of birds, looking for consistencies and differences.” to “For groups with sufficient sample sizes, we also used this evidence to assess variation in postures and body regions used by different individuals (3) within species, and (4) among taxonomic families of birds.”
- In Line 104, the placement of the Gill et al 2020 citation struck me as a bit odd, but it looks like you included that citation to identify the taxonomy you followed for species identification. In that case, you should say that, e.g. “…and 50 different families, following the IOC World Bird List taxonomy (Gill et al 2020).”
- Line 106-109 are redundant with the sections that follow.
- In the “Highlighted body regions” section, I think it is important that you define what “highlighted” means early in the paragraph (e.g. before you discuss grouping of regions).
- Line 123: what does “distinguished” mean here (and who is doing the distinguishing)? Do you mean that the authors of studies of coloration paid special attention to those regions, or that birds in those studies distinguished those regions somehow?
- In Line 138, does “focal regions of signaling” mean the same thing as “highlighted regions”? If so, please use consistent terminology. I prefer the “focal regions of signaling” terminology, since “highlighted” is passive and makes the actor (who is doing the highlighting, the bird or the authors?) unclear.
- Line 139, the use of the word “highlighted” is a bit unclear. I think you are saying that the displaying bird is “highlighting” regions that are particular colors, in the direction of the other individual. Please clarify: Is the protocol of your naïve scorers to examine a video/photo and identify three body parts that are prominently featured in the bird’s display and oriented toward another individual, and then to say the dominant color of the three body parts? Or do the naïve observers identify conspicuous colors/color patterns that are not necessarily on the three “highlighted” body parts?
- The sentence starting on line 185 should be the start of a new paragraph.
- In addition to the cases mentioned above, there are several sentences in the manuscript that use passive voice, which should be changed to active voice. For example, line 226 says “no one shoulder position was taken in the majority of aggressive signals…”
- Line 375 do you mean auditory signals, not visual?
- Word choice: line 379 you say your study highlights but also that there are highlighted regions. This goes back to my sense that “highlighted” regions or colors is a confusing word choice. In any case, the wording in this line should be altered. Also, I suggest using “differences” instead of “distinctions” in line 380.
- Line 326 needs a citation for bird-mammal signaling

Experimental design

The authors have clearly contextualized the work and addressed a knowledge gap. The data they collected are relevant for answering their outlined question and the analysis is done carefully. However, the paper requires additional details in the methods section. Furthermore, I have several significant questions and suggestions about the analysis.

Methods:
- Line 93 says that most of the videos were from WikiAves. That implies that other databases were used; please list these as well. In addition, please explain how you found videos/photos from these other databases (where did you search for them, what search terms did you use).
- Related to that, Figure 9 is a photo from Cornell bird cams (and is a great depiction of interspecific aggressive signaling!), which suggests that you might have analyzed some of the feeder camera footage? Is that the case? If so, please say that in the figure caption.
- Please include a short description of WikiAves for readers who are not familiar with the database (for example, the geographic range covered by this database, who contribute data to the site, what years of observations does the database cover).
- Please share more information about how you searched for photos/videos in WikiAves. There is an Advanced Search function in the database; did you use this, and if so, which search criteria did you use? (e.g., what did you select for “Photo content”, for “Main Action”? Did you examine all photos/videos that came up from this search or only particular dates?)
- In Line 103-104 (and later in the discussion), please also include the number of orders (not just genera and families).
- The “Postures,” “Highlighted body regions,” and “Highlighted colors” sections all refer to only interspecific interactions, and the “Relationship to intraspecific signals” section indicates that the photos/videos you analyzed were interspecific not intraspecific interactions. Therefore you should say in the first section, “Signal data”, that the photos/videos you collected were about interspecific interactions.
- That being said, I think the fact that some of the data on intraspecific signals were based on literature descriptions and thus collected in a different way than the interspecific signals info is a potential problem. The authors can address this by verifying that the video/photo approach yields comparable information to the descriptions in the literature. Please use the same protocol of examining/categorizing signaling in video/photos of intraspecific interactions for a subset of species and compare those findings to the intraspecific signaling descriptions from the literature.
- Lines 152-154: please say for how many species you used literature descriptions and for how many species you used videos/photos.

Analysis:
- It seems to me that your GLMMs should include signaling species and/or signaling family as a random effect. Also relevant to include as a random effect is whether the data come from a photo or a video (for example, it seems that an observer would be more likely to identify an open beak as important in a video where a bird is opening and closing its beak (since any opening of the beak during the clip would count as an open beak), as opposed to a photo where the image might have been taken right after the bird closed its beak).
- While the conditional inference trees (CIT) are an interesting way to present the data, I am not convinced that they are the appropriate analysis and visualization option here and would like the authors to provide some explanation about why they chose the method. I am familiar with classification and regression trees (CART), random forests, etc., and understand the difference in splitting criterion between CIT and CART. It seems to me, however, that CIT must suffer from one of the same limitations as CART, which is that the model is built by making univariate splits after linearly ranking predictor variables, when in fact the “true” relationship between predictors and the response could be nonlinear. As such, a classification trees (and perhaps CIT) suffer from much weaker predictive ability (especially when not pruned) than random forests, in exchange for being easier to interpret. So, I think the question here is—are you building a CIT to describe variation within and between families, or to build a model that generally captures important predictors of family-specific behavior (and thus want your model to do a good job of categorizing species)? If your goal is the latter, a random forest approach (using a CIT-type algorithm) would definitely be better. However, I think your goal is the first. I think your description in the results section of the CIT models you generated are consistent with this modeling approach (e.g., the most important variable is mouth, line 270), but in the discussion it sounds more like you are commenting on variation in and among families. For example, in the paragraph starting on line 327, you comment on certain postures or body regions used by certain families, as opposed to the CIT category which combines multiple postures or body regions. For this reason, I don’t think a CIT model is best. I feel that simply providing descriptive statistics and a visualization such as a grid of family x posture (or family x body region, or family x color) would be much easier to interpret and more in line with the goal of identifying commonalities and differences among families as well as variation within families. If you disagree with my assessment of the use of CIT, please provide more justification and explanation in the methods section.
- Line 201-202: You state that the goal of the CITs is to “examine how well taxonomic families could be categorized based on their postures,” but this wording is unclear (do you mean to say that you want to know whether different species in a family are placed into the same or different categories, or whether different taxonomic families could be grouped together?). Also, I think this stated goal follows the goal of CITs, but as described in the previous comment, I am not sure that this goal aligns with your previously stated goal of characterizing family-level variation in aggressive signaling (e.g., Line 86-87).
- If a certain family never highlighted certain colors (for example, carotenoids), is that because they have such coloration and don’t highlight it, or might it also be because they lack such coloration?

Validity of the findings

The authors have discussed the implications clearly without overstating the work. I found the points they brought up in the discussion to be interesting and well expressed. I have a few line comments for the discussion as well as some suggestions for additional topics to elaborate on in the discussion.

- Line 293 – I think you could cite only Fig 3. I have trouble easily gleaning this result from Fig 6 and 8.
- Line 294-295 Isn’t the result that the bill was oriented forward (either open or closed)? Or does forward mean close to the receiver?
- Discrepancy between sentence on line 338-339 and line 339-342, wording implies that black/white are carotenoid colors.
- Line 364-366: do you mean something like: “Importantly, we find that certain areas that have been omitted from studies of aggressive signaling, such as bright mouth linings and underwing patterns, are in fact frequently emphasized in the aggressive displays of many bird species.”

Suggestions for additional discussion:
- Discussion should discuss taxonomic/geographic limitations of your study. Do you expect any other kinds of signaling or any deviances from the patterns you found if you were to include other major groups of birds, or do you think these results can be broadly applied?
- Also, please discuss how aggressive signals might vary in other contexts. You didn’t examine cases where the birds ended up fighting or where one retreated; would you expect different signals to be used prior to those fights or in those situations?
- Any comments about WikiAves as a dataset? Or the value of citizen science databases like it?

Additional comments

Overall, I found the paper to be straightforward and a nice use of an interesting database to address a gap in our knowledge of interspecific aggressive interactions. I am confident that the authors will be able to address my comments.

I have a few remaining line comments:
- Figs 1 and 8: beautiful drawings! Please cite the artist (even if it’s one of the authors).
- Fig 3 and 4: can you jitter the gray points vertically or say the sample size at 1 or 0 instead? Currently it’s hard to get a sense for how many points are on the plot.
- Fig 5,6,7 remove node numbers so we focus on sample size numbers instead
- Small typo in the R file: you refer to the column signal.dat$signalling.families but in the dataset you provided, the column is called signaling.families.

---

## Round 0.2 · Major Revisions

· Academic Editor

Major Revisions

Thank you for the thoughtful responses to each reviewer. Both reviewers acknowledge these changes and considered the manuscript improved. However, both reviewers still have some outstanding comments that will require addressing, particularly related to the statistical methods used. I would be grateful if you were able to consider these suggestions in a revised manuscript.

Reviewer 1 ·

Basic reporting

No comment.

Experimental design

No comment.

Validity of the findings

Although the new analytical approach is undoubtedly more robust, there is still an issue with the PGLMMs (see Additional comments).

Additional comments

I appreciate the modifications made by the authors in response to my previous comments. They made substantial changes to address all of my major concerns. I am happy with the clarification about what I suspected could be circular reasoning in data collection and scoring. I was also glad to see that the authors rescored color data so as to make the color analysis more biologically relevant. The manuscript overall also reads more focused and streamlined in this version, in part because one of the previous analyses was appropriately removed. Finally, the new, phylogenetically controlled analytical approach provides much more robustness to the authors’ findings (but see caveat below). Overall, I find the manuscript in its new version to be a valuable contribution to the avian aggressive signaling literature.

I have a few minor comments that I will outline below.

L153: “We defined highlighted regions as those that were more obviously displayed or modified for display to the receiver” – seeing as naïve observers, not the authors, were responsible for scoring highlighted regions, it seems to me that the definition should be whatever was passed as instructions to the observers. No definition was provided to observers according to Appendix II, in which case I would suggest that this sentence (L153–155) be excluded. The next sentence (“perceived as most highlighted [by observers]”) more accurately describes the scoring procedure.

L158 and 169: reference to Appendix II would help readers.

L215: I would hate to have the authors conduct these analyses once again, but, as I had pointed out in my previous review, the best practice for PGLMMs with multiple measurements is to include species identity as a random effect, *in addition* to phylogeny. See http://www.mpcm-evolution.com/practice/online-practical-material-chapter-11/chapter-11-2-multiple-measurements-model-mcmcglmm.

L355–371: I like this paragraph relating aggressive displays to weapons. Perhaps the authors could also comment on the species of birds that possess specialized weapons (bony spurs) on the wings or leg, two of which are in the dataset (Vanellus and Jacana; also Gallus but not hens)? Spurs appear to be highlighted by both species in the assessed pictures.

By the way, wing pounding is not unique to pigeons and doves (L370). Besides Columbiformes (e.g., account by Leguat 1708 in Hume & Steel 2013, https://doi.org/10.1111/bij.12087), species in Anseriformes (e.g., Nuechterlein & Storer paper cited in the manuscript) and Charadriiformes (e.g., Walters 1979, https://doi.org/10.1016/0003-3472(79)90045-9) that possess wing spurs use them for wing pounding.

L374: Body size is also a good predictor of winner of fights across animals (Palaoro & Peixoto 2021, https://doi.org/10.1101/2020.08.26.268185).

L452–459: I really liked the authors’ decision to include this paragraph in the main text rather than in the acknowledgments as would be expected. I was glad to see this invaluable resource from my home country duly emphasized.

Figure 2 looks so much nicer now, with the estimates and 95% CI instead of the bar plots. Two minor suggestions: sort panels (c) and (h) like the other panels; indicate in panel (c) what positions were grouped as “open wings”.

Figures 4 and 5: I suggest reversing the time axis (i.e., starting at ~120). The axis would then be interpretable as million years before present (Mybp) or million years ago (Mya) — more intuitive to readers than branch length.

Love new Figure 5 — very informative! I would consider using black-and-white striped pattern for dark/white contrast and white tile with black borders for white (not sure if the latter will work well, but I think it is worth trying). Also, I think the authors meant “brown for brown.beige” in the legend.

Reviewer 2 ·

Basic reporting

The authors have addressed many of my concerns and have greatly improved the manuscript. However, I still found parts of the manuscript difficult to understand and think that the authors should revise to improve clarity, especially in the methods. I also have major concerns about the choice of statistical analyses.

- Lines 104-109: Please clarify this wording. From these two sentences, it’s unclear whether the data from the Internet Bird Collection ended up being used or not, given that you did not include data from the Macaulay Library. Do you mean that OTHER data from the Macaulay Library (not the IBC data that you had searched before it was moved to the Macaulay Library) you didn’t use, but you did use IBC data? It’s unclear when you say “we omitted this data source from our study” which data source you are referring to.
- On the topic of the choice of the word “highlighted”, I agree with the authors that it is important to use that terminology given that that was the term they used in instructions to data collectors. However, please avoid using the passive voice (e.g., “were highlighted” in Line 136) when you do use this word, to improve clarity.
- Lines 210-212 make it sound like the postures/colors analyses were done for intra vs interspecific comparisons. Consider rephrasing to something like: “We used phylogenetic mixed-effects models using the MCMCglmm function in the MCMCglmm R package (version 2.32; Hadfield 2010) to test whether birds were more likely to use specific postures or highlight specific body regions or colors in aggressive encounters with heterospecifics, as well as to compare whether birds use the same signals in intraspecific aggressive interactions as in interspecific aggressive encounters.”
- I’m very confused by 244-249 wording. What does it mean for a bird to assume a degree of similarity?
- Line 259 suggested wording change: “we calculated the proportions”
- Figure 2 and Figure 3b, 3c are low quality/blurry and difficult to read. Please improve the quality and make the text larger/points darker to improve legibility.

Experimental design

- The wording of line 189-191 is confusing. First, I don’t think you mean that species are uninformative, so you should write something like: “We did not incorporate color into these comparisons because some species have uniform coloration and thus color is uninformative with respect to patterns of within- versus among-species signaling”. Second, it doesn’t make sense to ask if a bird “highlights” a color when it has a uniform color/is only that color, regardless of whether you’re doing within- or between-species comparisons. Please clarify if you are including birds that are only one color in your color analyses. Do you instead here mean “…some species have no/low within-species color variation…?”

Validity of the findings

- The interpretation of the glms starting on Line 270 is very confusing. I had trouble matching the p-values reported in the text with the values reported in the supplemental tables. I also don’t agree that it is okay to interpret an intercept with a p-value > 0.5 as being a significant driver of the outcome just because the other predictors in the model are significantly lower than the intercept. Most importantly, however, I do not think that models in the GLMM family are appropriate for this analysis (and for some of your other analyses). If your question is whether or not the different options for a given posture, e.g. body position, are equally likely, I think you want a model in the chi-square family. This same critique can be applied to many of your analyses. I believe that the authors need to re-examine and simplify many of their analyses and work on the description of their analysis to make what they did very clear to the reader. If I have misunderstood, please explain and justify your methods better.
- In your supplemental tables, please label the p-value as pMCMC, and in the table caption clarify that the 95% CI are credible intervals.

---

## Round 0.3 · Minor Revisions

· Academic Editor

Minor Revisions

Thank you very much for your resubmission. As you can see, Reviewer 1 raises a concern related to a minor issue in the code for the analyses. I think that this is a very helpful observation, and it appears that it will be relatively straightforward to deal with. It seems that it is important to correct this before publication. This is the only change that is required in this final round of revisions.

Reviewer 1 ·

Basic reporting

See below.

Experimental design

See below.

Validity of the findings

See below.

Additional comments

I appreciate the authors addressing the few comments that I had made on my previous report and maintain that this manuscript would be a valuable contribution to the aggressive signaling literature.

I do have to say that, upon inspecting the codes, I came across issues that I had failed to notice in my previous revision. I apologize for bringing up new topics in the third round of revision but hope that the authors understand that refraining from doing so would mean not serving appropriately in my role as a reviewer. Most importantly, however, I trust that these issues (a) will not qualitatively (and barely quantitatively) affect the results and (b) will not be hard to deal with.

The first is what is likely a typo in the codes. All priors were defined as lists containing the element “n”, when MCMCglmm needs the element “nu”. Simply replacing all instances where “n=k” occur with “nu=k” solves this issue. I am not completely sure that this is indeed a typo — perhaps it is an issue that arose in the new version of MCMCglmm (the authors used v2.32 for analyses, while I am testing their codes with v2.33).

The second is that the species Torgos tracheliotos appears with two different IOC codes (signaling.IOC) in the data set, causing this species to behave differently in relation to the random factors “animal” (such that they are considered to be on the same tip of the phylogeny) and “signaling.IOC” (such that they are treated as distinct species). I corrected this issue (i.e., made it so that all Torgos tracheliotos have the same signaling.IOC) and reran all models with much fewer iterations as a test. Results changed only very slightly (if at all). I am pretty sure that correcting this issue will barely change the results quantitatively (certainly not change them qualitatively) and so will leave it to the editor’s and authors’ discretion to decide on whether it is worth the trouble of rerunning analyses.

The third is related to the authors’ choice of using no thinning interval for the MCMC chains. Using no thinning with traditional MCMC samplers like the ones MCMCglmm uses (as opposed to more efficient samplers like HMC or NUTS) causes a very high autocorrelation between successive samples — this can be verified with the function coda::autocorr. Excessive autocorrelation is not a problem in itself but it invariably leads to very reduced effective sample sizes (ESS). That is why ESS rarely exceeds 30,000 in Table S12, even though each chain sampled 990,000 estimates per parameter. The authors mention in the Supporting Information that an ESS of 200 is satisfactory*. I could not find this information in the references provided and would appreciate if the authors could point me to it, because I seem to remember seeing higher cutoffs generally being used as a rule of thumb (at least 1000, if I am not mistaken). Not all parameters would pass a 1000 cutoff, especially for the model in Table S12i. For future reference, I would recommend that the authors use a thinning interval of at least 100 — this will improve the ESS-to-total sample ratio and spend less memory on redundant samples. But for the purpose of this study, there is an easier way around that does not require rerunning models.

[*By the way, the authors mention that an ESS > 200 denotes chain convergence, but I find this statement to be strange because, as far as I know, ESS provides no information on chain convergence. ESS is simply an estimate of how large the sample would be if estimates were independent from each other. This information can indicate how efficient the sampler is, not whether chains converged well. Convergence, again as far as I am aware, is instead tested by comparing the behavior of multiple chains, like Gelman–Rubin diagnostics do.]

Since the authors ran three chains per model to test convergence, and they found that the chains did converge, the solution is simple: they can safely combine the estimates from all three chains (which they already have in hand — no need to rerun) and thereby increase ESS by a factor of approximately three. Regardless of cutoffs, a higher ESS will always provide more robust estimates, and that is especially true for Table S12i. To implement what I am suggesting, all the authors have to do is combine the chains before computing ESS, mean, and 95% CI. I will provide an example for the body orientation model below — hopefully it will be of help for the other models as well.

# combining fixed effect estimates as a list of matrices
Sol <- mcmc.list(body.orientation.model.MCMCglmm$Sol, body.orientation.model.MCMCglmm.2$Sol, body.orientation.model.MCMCglmm.3$Sol)
# extracting ESS
round(sort(effectiveSize(Sol)))

# combining random effect estimates
vv <- mcmc.list(body.orientation.model.MCMCglmm$VCV, body.orientation.model.MCMCglmm.2$VCV, body.orientation.model.MCMCglmm.3$VCV)
# extracting ESS
effectiveSize(vv)

# combining fixed effect estimates as a single matrix
est <- rbind(body.orientation.model.MCMCglmm$Sol, body.orientation.model.MCMCglmm.2$Sol, body.orientation.model.MCMCglmm.3$Sol)
# extracting mean for forward.lowered
forward.lowered.est<-inv.logit(mean(est[,"(Intercept)"]))
# extracting 95% CI for forward.lowered
forward.lowered.ci<-inv.logit(HPDinterval(mcmc(est[,"(Intercept)"])))

Finally, a small correction regarding L140: spurs in the mentioned species are located in the carpal joint (i.e., the wrist, not the elbow).

---

## Round 0.4 · accepted · Accept

· Academic Editor

Accept

Thank you for your revised manuscript and attention to the reviews. Congratulations on such an interesting paper.